

# Application of Graph Theory to the elaboration of personal genomic data for genealogical research

Vincenzo Palleschi[1,2], Luca Pagani[3,4], Stefano Pagnotta[1], Giuseppe Amato[5] and Sergio Tofanelli[6]

[1] Institute of Chemistry of Organometallic Compounds, Research Area of National Research Council, Pisa, Italy
[2] Department of Civilizations and Forms of Knowledge, University of Pisa, Pisa, Italy
[3] Division of Biological Anthropology, University of Cambridge, Cambridge, United Kingdom
[4] Department of Biological, Geological and Environmental Sciences, University of Bologna, Bologna, Italy
[5] Institute of Sciences and Technology of Information, National Research Council, Pisa, Italy
[6] Department of Biology, University of Pisa, Pisa, Italy

## ABSTRACT

In this communication a representation of the links between DNA-relatives based on Graph Theory is applied to the analysis of personal genomic data to obtain genealogical information. The method is tested on both simulated and real data and its applicability to the field of genealogical research is discussed. We envisage the proposed approach as a valid tool for a streamlined application to the publicly available data generated by many online personal genomic companies. In this way, anonymized matrices of pairwise genome sharing counts can help to improve the retrieval of genetic relationships between customers who provide explicit consent to the treatment of their data.

## INTRODUCTION

In recent years, a number of companies started offering commercial services based on DNA analysis for genealogical research (https://genographic.nationalgeographic com/, https://www.23andme.com/, https://www.familytreedna.com/). The informatic tools available to interpret such results, usually provided by the same companies or by external services (http://www.gedmatch.com/), are mainly focused on general population studies (Paternal and Maternal lineages based on Y chromosome and mitochondrial haplogroups, Ancestry Composition/Admixture, etc.). On the other hand, very few tools are provided to investigate the links of one's DNA profile with the relatives made recognizable through personal genomic data. Notably, these pre-compiled tools are often the only way to access the data provided by the DNA testing companies for a panel of hundreds or thousands of individuals. Therefore, the starting point of any downstream analysis based on this kind of data can only rely on the semi-processed input provided by the aforementioned tools. The introduction by the genetic service providers of a wrapped application tool would facilitate users' interpretations and unearth hidden genealogical information. Such tool should

Corresponding author
Luca Pagani,
lp.lucapagani@gmail.com

enable to implement the mass of data each single DNA test makes available in an easy-to-grasp graphical form. This would be particularly useful to detect the provenience of distant autosomic DNA-relatives from either the paternal or the maternal lineage. In fact this task is often made difficult by the links that might exist between the two parental genealogies due to the custom in closed communities to marry between relatives, especially in the past.

Here we describe and annotate an artificial intelligence tool that helps exploiting the information provided to customers by genealogical genetic services. The original approach of this work is the use of cross-information about the links between the living DNA-relatives of the test user (TU) for obtaining hints about the possible connections with other individuals, in the absence of a-priori genetic or genealogic evidence.

## DATA

We performed quality checks via using the theoretical amount of genome sharing (Table S1) to simulate a similarity matrix (Table S2) based on two identical, hypothetical genealogies each made by 10 samples (individuals 1–10 and 11–20, Fig. S1). Pairs including individuals not linked in the genealogy were given a random amount of genomic sharing comprised between 0 and 2 Mbp.

We then used actual genomic data, consensually provided and anonymously treated, and derived from the results obtained by a test-user (TU) from the personal genomic service 23andMe (https://www.23andme.com/). Such results typically consist of summary statistics on about one million single nucleotide polymorphisms (SNPs) (*Nachman, 2001*).

A total of 120 anonymized individuals (progressively numbered with an ID from 1 to 120) were considered in the analyses. All of them are 'DNA-relatives' of the TU according to the 23andMe criteria and accepted the invitation to share their DNA information (excluding data related to health conditions). The raw data is available in Table S3. Since this is a secondary analysis of pre-existing data and the samples are treated in an anonymised version we did not apply for an ethical clearance.

We also retrieved an additional genome sharing matrix from an independent test user (TU2, Table S4) who agreed to donate the matrix s/he obtained from 23andMe to be processed anonymously, in compliance with the ethical considerations provided in the paragraph above. This second matrix was used solely as a mean of independent confirmation of the validity of the approach presented here.

As reference parameter we considered the total amount of autosomal DNA in common between pairs of individuals, calculated as the total length of shared SNP haplotype blocks in mega base-pairs (Mbp) units. This amount, once converted into proportion of shared genome, provides a rough estimate of the number of generations separating any two individuals, under a simple model of "infinite number of ancestors" (Table S1). Information either on the relevant chromosomes where the match occurs, or on the number of segments in common was not used. This choice is justified by the fact that only a minimum percentage of the individuals considered shows DNA matches on more than one chromosome. Furthermore, the information about the specific segment of the chromosome where such match occurs is not easily obtainable from the data made available to the users by 23andMe.

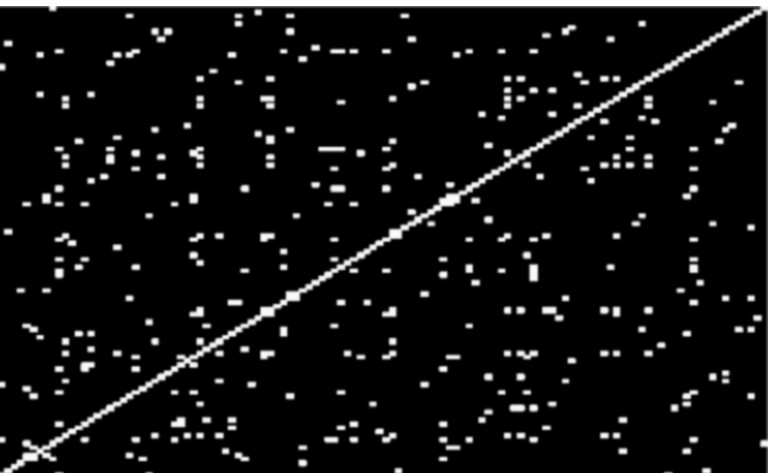

**Figure 1 Visual representation of the correlation between the individuals considered in this work.**

Using the Genome-Wide Comparison option in the 23andMe 'Family Traits' feature, the input data were prepared in the form of a symmetric square matrix $C$, whose $C(i, j)$ elements correspond to the total length of shared SNP haplotype blocks between the individual $i$ and the individual $j$, expressed in Mbp units. Most elements of the matrix are equal to zero, corresponding to the fact that the majority of the individuals does not result genetically related. The sparsity of the matrix $C(i, j)$ is visually shown in Fig. 1, where the white points indicate a mutual match of any magnitude between two individuals, and the black ones correspond to no genetic relation at all.

## CLASSIFICATION

The matrix (Table S3) depicted in Fig. 1 can be alternatively interpreted as a correlation matrix, a covariance matrix, a similarity matrix (*Srivastava, 2002*) or it can be transformed in a distance matrix (*Smouse & Long, 1992*) Accordingly, the way to elaborate and manipulate the associated information varies depending on the interpretation tasks. Given that the statistical analysis is aimed at simplifying data outputs, a loss of information with respect to the original data has to be expected. The effectiveness of the analysis thus depends on the amount of 'interesting' information unearthed out of the bulk of 'redundant' information. It follows that different methods can be more or less effective according to what is considered, from time to time, interesting or redundant.

To this extent, a number of potential confounders must be considered when dealing with the available genetic similarity matrix. First of all the genetic information on which the analysis is based is intrinsically fuzzy, because of the uncertainty in the data obtained by the service provider (a few 'no-called' SNPs should be routinely expected). Additionally, the presence of identical by state (IBS) other than identical by descent (IBD) (*Stevens et al., 2011*) SNPs could potentially bias the genealogical interpretation, especially the one associated with distant relationships (Most Recent Common Ancestors distant more than 6/7 generations). Finally, as opposed to uniparental markers, the diploid autosomic data combine information inherited from the paternal and maternal genealogy that should be

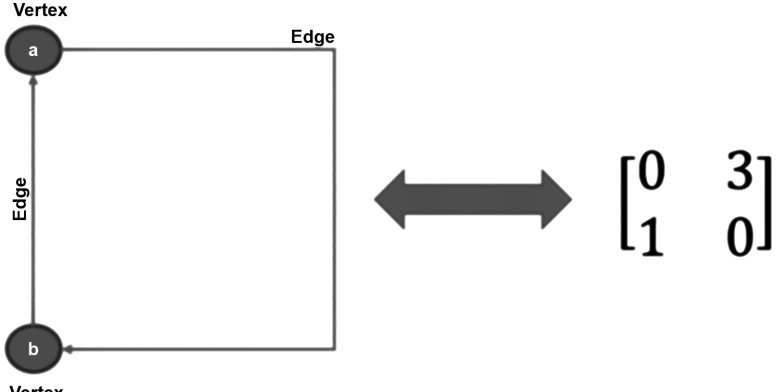

**Figure 2** **Graphic representation of a Graph with two vertexes and two edges (oriented Graph).** On the right, the corresponding adjacency matrix.

kept separated when tracing one's ancestry. Therefore, the analysis must be performed using statistical techniques robust enough to sustain these unavoidable uncertainties.

## GRAPH THEORY APPROACH

The ideal framework for studying the complex network of links between the DNA-relatives of a TU is the Graph Theory (*Bondy & Murty, 2008*; *Pavlopoulos et al., 2011*). This approach, widely used in Mathematics, Engineering, and Computer Science, allows the analysis and graphical representation of the links between different entities in a network. In synthesis, the Graph Theory represents the elements in a network as *vertices* (or nodes) connected by *edges*. Edges are often associated with a value representing a *weight*. In our case, the weight of an edge connecting two vertexes is related to the genetic distance between them. A couple of vertexes *a* and *b* can be connected, in principle, by more than one edge. Graphs can be generally oriented, so that the edge from *a* to *b* is different from that linking *b* to *a*. In this way, the distance between the vertexes *a* and *b* can be different from the distance between *b* and *a* (a typical example is driving a car between two points in a city, where the traffic regulations might impose different routes for the direct and return trip, see Fig. 2).

The relation between the vertexes is usually represented in matrix form (adjacency matrix *Gehlenborg & Wong, 2012*) where the elements out of the diagonal are the weights of the corresponding edges. If the adjacency matrix is symmetric (the distance between two nodes is the same in both directions) the resulting graph is called *unoriented*.

In our scenario, the correlation matrix $C(i, j)$ between the DNA-relatives of the TU is interpreted as a symmetric adjacency matrix. Therefore, we will use unoriented graphs, implemented using the Matlab[®] code provided in Supplemental Information.

## RESULTS

The preliminary runs we performed on the simulated data (Table S2 and Figs. S2–S7) showed that, once the effect of the randomly introduced noise was taken into account, the Graph approach yielded the expected inter-individual relationships. Particularly, the introduction of a 3 Mbp cutoff (Fig. S4) to remove genetic links arising from the randomly

added genetic similarities (≤2 Mbp) managed to re-create the simulated scenario (Fig. S1). More stringent thresholds (up to 48 Mbp, Figs. S5–S7) further simplified the picture, leaving as viable connections only the individuals with the closest relationships. Remarkably, the two simulated genealogies (individuals 1–10 and individuals 11–20) were treated in the exact same manner by our approach, hence showing its robustness. This exercise served as a proof of principle to show that the introduction of a cutoff to remove genetic links below a certain threshold is beneficial to the removal of noise. According to the obtained results, a cutoff between 3 and 6 Mbp (Figs. S4 and S5) is sufficient to remove noise while keeping in the pictures genetic links up to seven generations. It follows that in situations where the available genetic data is made up only by distantly related individuals (i.e., more than seven generations), a cutoff between 3 and 6 Mbp is sufficient to remove the background noise, while keeping the true genealogical information embedded in the data.

We then applied the Graph Theory approach to the empirical genomic data. In the dataset analysed here, the TU adjacency matrix (Table S3) is described by an unweighted Graph with 120 vertexes (individuals) and 196 edges (DNA links between them). The graphical representation of the Graph described by this matrix is shown in Fig. S8.

The main network connects 100 vertexes (83% of the total) by 190 edges (97% of the total) and sets aside only a few individuals, singularly (10 individuals) or in small groups of two or three persons. A strict interpretation of Fig. S8 would thus bring to the conclusion that all the individuals belonging to the main group should be considered as somehow related, directly or indirectly, to all the other members of the group. To reduce this connectivity and to assign the various individuals to the TU paternal and maternal ancestries, a further treatment of the input data is thus necessary.

## Pruning

As shown for the simulated genealogies, the strength of the DNA cross-links between the individuals can be used to reduce (prune) the connections highlighted in Fig. S8. Since all the 120 individuals included in this study are, by design, related with the TU, no information can be derived from those that are connected only to the TU. They are represented, in graphical form, as isolated vertexes with no edges associated. Therefore, these individuals can be safely removed from the adjacency matrix without any loss of information. Moreover, as already discussed in 'Classification', spurious connections could be introduced by fuzziness of the genetic data and the occurrence of IBS SNPs. These connections can be excluded via the application of an upper threshold on the genetic distances between the individuals. The threshold amount of shared genome for a link to be considered 'real' (i.e., corresponding to IBD SNPs) can be easily converted into expected number of generations, using Table S1.

Following the results on the simulated data, and given the abundance of strong genetic links within the TU similarity matrix, we chose to apply a more stringent threshold to increase the readability of the resulting Graph. Figure 3 shows the Graph where only the edge weights greater or equal to 24 Mbp (roughly a 8 generations distance between the vertexes/individuals, see Table S1) are considered.

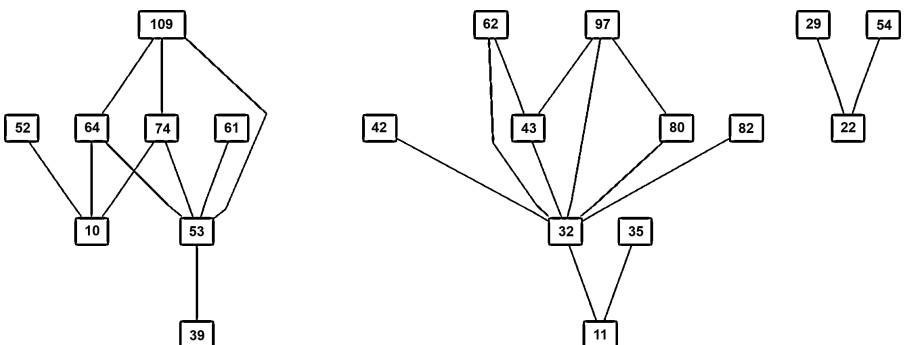

**Figure 3** **Graphic representation of the Graph described by the adjacency matrix** $C(i, j)$ **considering only the edges corresponding to DNA-matches greater or equal to 24 Mbp.** Isolated individuals and groups of two are not reported in the figure.

**Table 1** **Classification of the individuals according to their lineage (24 Mbp threshold).** Individuals underlined and marked in bold are the ones for whom a genealogical evidence exists, and therefore can be assigned with certainty to a given lineage. The ones underlined and marked in italic, on the other hand, cannot be assigned with similar certainty, although there are strong independent clues suggesting that they would actually belong to that lineage.

| Paternal GF | Paternal GM | Maternal GF | Maternal GM | Unclassified |
|---|---|---|---|---|
| | | 62 | 52 | 28 |
| | | **97** | *109* | 54 |
| | | **42** | 84 | *22* |
| | | 43 | *74* | |
| | | 80 | 61 | |
| | | 82 | *10* | |
| | | **32** | *53* | |
| | | **35** | 39 | |
| | | **11** | | |

Figure 3 corresponds to the idea of unconnected graph that we associate with the separation of the different ancestral lines of the TU. Surprisingly enough, when the results of the Graph Theory are compared with the pre-existing genealogical information on some of the matching individuals, it turns out that the two large groups correspond to relatives of the TU related to the maternal grandfather (at the center of the figure) and maternal grandmother (to the left). Another small group of three individuals, to the right in Fig. 3, shows up, containing an individual associated to the maternal grandmother's lineage of the TU (n.22). The two individuals that can be identified with reasonable certainty as belonging to the paternal grandfather's (n. 118) and grandmother's (n. 96) lineage of the TU, remains unconnected. These results are summarized in Table 1. The individuals underlined and marked in bold are the ones for whom a genealogical evidence exists, and therefore can be assigned with certainty to a given lineage. The ones underlined and marked in italic, on the other hand, cannot be assigned with similar certainty, although there are strong independent clues suggesting that they would actually belong to that lineage.

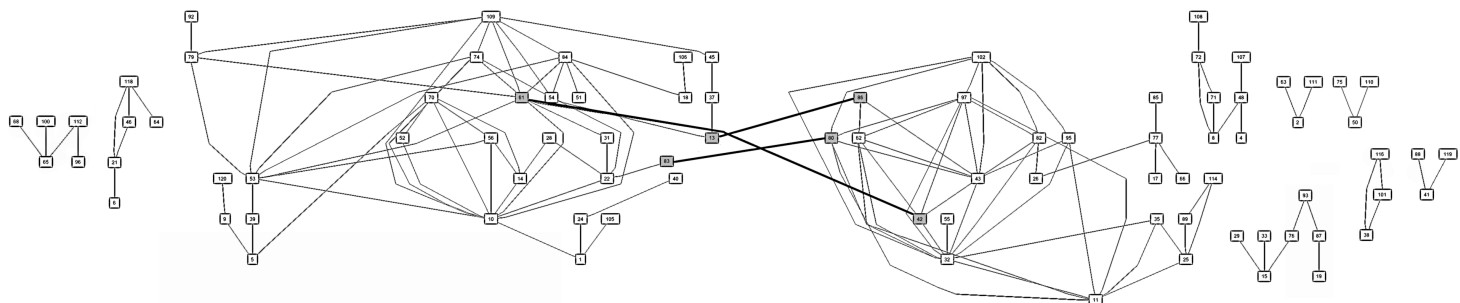

**Figure 4 Graphic representation of the Graph described by the adjacency matrix $C(i, j)$ considering only the edges corresponding to DNA-matches greater or equal to 6 Mbp.** Isolated individuals and groups of two are not reported in the figure.

The adoption of a conservative threshold (24 Mbp/approx. 8 generations distance/3rd–4th cousin range) to define a link between the individuals produced the classification reported in Table 1, which is robust and reliable. However, only 17 individuals over a total of 120 (110 with at least one DNA match besides the TU) are attributed to the corresponding ancestral lineage.

Reducing the level of the threshold to 12 Mbp (approx. 9 generations distance) increases the number of individuals that can be associated to the different groups (Fig. S9). Individual 22 is now correctly associated to the maternal grandmother's group, along with the other members of his/her subgroup. Most importantly the graph now shows an additional group of three individuals (21, 46 and 118) that can be associated to the TU paternal grandfather's lineage, on the basis of independent genealogical information existing for individual 118.

Further lowering the threshold to 6 Mbp, the threshold tested on the simulations (approx. 10 generations distance, i.e., a 4th–5th cousin range, which is usually considered the lower limit for having a significant DNA match between two individuals) allows to recover important information, graphically represented in Fig. 4.

From the analysis of Fig. 4 it is evident that after lowering the threshold to 6 Mbp, a connection appears between the two main groups. The key elements which are linked to both groups (corresponding to the maternal grandparents of the TU) are individual 83 (initially classified in the maternal GM group) which connects with individual 80 in the maternal GF group, individual 61 of the maternal GM group which connects with individual 42 in the maternal GF group, and individual 86 of the maternal GF group which connects with individual 13 of the maternal GM group.

Lowering the threshold also increased the number of individuals associated to the paternal grandfather of the TU, which at this level formed a group of five persons (118, 46, 21, 6 and 64) connected by the same sub-graph, and recovered a new group of five individuals (96, 112, 65, 100 and 68) that can be associated to the TU paternal grandmother's lineage on the basis of independent genealogical information existing for individual 96.

The main information that can be derived by the comparison of the Graphs obtained using different thresholds on the edge weight is a classification of the individuals according to the different ancestral lineages, with increasing 'levels of confidence'. In that respect,

Table 2 **Classification of the individuals according to their ancestral lineage.** The corresponding level of confidence of the classification is reported in brackets. The individuals connecting the two groups of the maternal grandparents are marked in gray.

| Paternal GF | Paternal GM | Maternal GF | Maternal GM | Unclassified |
|---|---|---|---|---|
| *118*(12) | *96* (6) | **97**(24) | 52 (24) | 116 |
| 46 (12) | 112 (6) | 62 (24) | *109* (24) | 101 |
| 21 (12) | 65 (6) | **42** (24) | 84 (24) | 38 |
| 6 (6) | 100 (6) | 43 (24) | *74* (24) | – |
| 64 (6) | 68 (6) | 80 (24) | 61 (24) | 29 |
|  |  | 82 (24) | *10* (24) | 33 |
|  |  | **32**(24) | *53* (24) | 15 |
|  |  | **35** (24) | 39 (24) | 76 |
|  |  | **11**(24) | *70* (12) | 93 |
|  |  | 102 (12) | 83 (12) | 87 |
|  |  | 95 (12) | 54 (12) | 19 |
|  |  | 25 (6) | 51 (12) | – |
|  |  | 89 (6) | 92 (12) | 63 |
|  |  | 114 (6) | 79 (12) | 111 |
|  |  | 83 (6) | *14* (12) | 2 |
|  |  | 61 (6) | *22* (12) | – |
|  |  | 85 (6) | 9 (12) | 75 |
|  |  | 77 (6) | 5 (12) | 110 |
|  |  | 17 (6) | 28 (12) | 50 |
|  |  | 66 (6) | 1 (6) | – |
|  |  | 26 (6) | 24 (6) | 88 |
|  |  | 55 (6) | 56 (6) | 119 |
|  |  | 13 (6) | 42 (6) | 41 |
|  |  | 86 (6) | 18 (6) | – |
|  |  |  | 106 (6) | 108 |
|  |  |  | 31 (6) | 72 |
|  |  |  | 13 (6) | 71 |
|  |  |  | 86 (6) | 8 |
|  |  |  | 45 (6) | 48 |
|  |  |  | 37 (6) | 4 |
|  |  |  | 120 (6) | 107 |
|  |  |  | 105 (6) |  |
|  |  |  | 40 (6) |  |
|  |  |  | 80 (6) |  |

Fig. S8 would give a minimum level of information, providing classification at the confidence level of the minimum match in the $C(i, j)$ matrix, which in our case is 2 Mbp, subsequently refined at higher thresholds of genomic sharing in Figs. 3, S9 and 4.

The most important results of this paper are shown in Table 2, where the classification of the DNA-relatives of the TU is reported according to his maternal and paternal ancestral lineages, with the corresponding confidence level, or 'strength', in brackets. The individuals

connecting the groups corresponding to the two maternal grandparents are assigned to both the groups and marked in gray.

The Graph Theory method here proposed is capable of reliably classifying 62 individuals at strength 6 (Mbp) over a total of 110 DNA-relatives of the TU (56%). Six other unclassified groups with more than two members can also be determined. Some of them could be connected to the main groups if additional information from new DNA relatives of the TU will become available in the future.

We further validate our approach on an additional genomic matrix (TU2), without (Fig. S10) and with the 6 Mbp threshold (Fig. S11). The Graph obtained without the threshold (Fig. S10) notably includes link with the "Mendel family", a real genealogy made freely available by 23andMe after assigning it a mock family name. Given the lack of known relationship between TU2 and the Mendel family we take the existing link as further support for the need of a 6 Mbp threshold when interpreting the genetic results. The application of such a threshold (Fig. S11) indeed yields a cleaner Graph, with marked separations between the putative paternal and maternal TU2 family branches.

## CONCLUSION

The statistical method presented in this work can be usefully exploited for extracting genealogical information from genetic/genomic data. The input data are usually 'fuzzy' and, therefore, the methods used for their analysis should be robust enough for providing useful information. The approach proposed, based on the Graph representation of the adjacency matrix built from the mutual matches between the DNA-relatives of the test user, after the setting of a 6 Mbp threshold fulfils this requirement. The method, for which the code is provided at the bottom of this paper, could be easily implementable by the genetic service providers for an easy visualization of the DNA-links existing between the customer and the other users of the service, at different levels of confidence.

### Funding
The authors received no funding for this work.

### Competing Interests
The authors declare there are no competing interests.

### Author Contributions
- Vincenzo Palleschi conceived and designed the experiments, performed the experiments, analyzed the data, wrote the paper, prepared figures and/or tables, performed the computation work, reviewed drafts of the paper.
- Luca Pagani conceived and designed the experiments, wrote the paper, prepared figures and/or tables, reviewed drafts of the paper.

- Stefano Pagnotta and Giuseppe Amato performed the experiments, analyzed the data, contributed reagents/materials/analysis tools, performed the computation work, reviewed drafts of the paper.
- Sergio Tofanelli conceived and designed the experiments, contributed reagents/materials/analysis tools, performed the computation work, reviewed drafts of the paper.

## Data Availability

23andme: https://www.23andme.com/you/relfinder/ (requires account).

## Supplemental Information

Supplemental information for this article can be found online at http://dx.doi.org/10.7717/peerj-cs.27#supplemental-information.

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
