# Peer review of "Application of Graph Theory to the elaboration of personal genomic data for genealogical research"

_PeerJ Computer Science, doi:10.7717/peerj-cs.27_

## Round 0.1 · original submission · Minor Revisions

Following the review of your Research Article, I recommend that it should be revised taking into account the changes requested by the reviewer(s).

Reviewer 1 ·

Basic reporting

Supplementary Table 1 is vital for conclusion of this paper. But there is no description of how to derive this table within the paper. Could the author explain more about it?

Experimental design

Is this a robust method to analyze another dataset? Multiple testing data is required.

Validity of the findings

1. How does the proposed method deal with potential confounders? data fuzziness and dentical by state (IBS).
2. The design itself make sure that the TU is linked to all the other individuals. But is it true from the observed data? Does TU have shared haplotype larger than 2Mbp with everyone?
3. Could the author compare predictions of the proposed method to that of Y chromosome based method? At least they can compare the paternal grandfather lineage predictions.

Additional comments

This paper is target specific, concise and clear. It solves an interesting problem with simple Graph Theory. However, the author should do some comparison with previous methods to emphasize its unique advantages and apply it onto multiple datasets to prove its general applicability.

Reviewer 2 ·

Basic reporting

The paper proposed a method to extract genealogical information at different levels by adjusting the threshold of shared genome. The paper presented an interesting topic, but failed to provide and compare with state-of-art research and advancement in genealogical study.

Section 3 claims that the analysis must be performed using statistical techniques robust enough to sustain unavoidable uncertainties. But the rest of paper does not apply or prove such techniques.

Experimental design

The analysis starts from 8 generation distance, and then loose the condition to 9 and 10 generation distances. Why choose 8 as the starting point? Is this choice applicable to all cases?

Validity of the findings

No Comments.

Additional comments

The method used in the paper is threshold-based filtering on similarity matrix, without serious links to graph theory.

---

## Round 0.2 · Minor Revisions

As you can see, Reviewer 1 has serious concerns about data availability. Please address this concern.

In addition, I would like to suggest that the authors add more discussion about the thresholds.

Reviewer 1 ·

Basic reporting

No Comments

Experimental design

No Comments

Validity of the findings

Due to the unavailability of data, the author can not independently validate their methods. This is the major concern.

The choice of threshold on the links between the different individuals is very important. But there is no clear standard to choose it.

Additional comments

No Comments

Reviewer 2 ·

Basic reporting

The majority of my concerns have been addressed except the use of term Graph Theory. However, this is a debatable issue and should not affect the acceptance.

Experimental design

No Comments

Validity of the findings

No Comments

---

## Round 0.3 · accepted · Accept

Thank you for your submission to PeerJ computer science. Please prepare high quality figures according to the PeerJ Computer Science instructions.